# Effects of Drought Stress on Photosynthesis and Chlorophyll Fluorescence in Blue Honeysuckle

**DOI:** 10.3390/plants13152115

**Published:** 2024-07-30

**Authors:** Weijiao Yan, Yongchuan Lu, Liangchuan Guo, Yan Liu, Mingkai Li, Boyuan Zhang, Bingxiu Zhang, Lijun Zhang, Dong Qin, Junwei Huo

**Affiliations:** 1Key Laboratory of Biology and Genetic Improvement of Horticultural Crops (Northeast Region), Ministry of Agriculture and Rural Affairs, College of Horticulture & Landscape Architecture, Northeast Agricultural University, Harbin 150030, China; 15136565478@139.com (W.Y.); 15962220660@139.com (Y.L.); b220401007@neau.edu.cn (L.G.); liuyanooo11@163.com (Y.L.); 15163423599@139.com (M.L.); z13251668068@163.com (B.Z.); dongq9876@126.com (D.Q.); 2National-Local Joint Engineering Research Center for Development and Utilization of Small Fruits in Cold Regions, Northeast Agricultural University, Harbin 150030, China; 3Heilongjiang Institute of Green Food Science, Harbin 150000, China; zlj852588@163.com

**Keywords:** blue honeysuckle, drought stress, chlorophyll content, photosynthetic activity, chlorophyll fluorescence parameters

## Abstract

Blue honeysuckle (*Lonicera caerulea* L.) is a deciduous shrub with perennial rootstock found in China. The objectives of this study were to explore the drought tolerance of blue honeysuckle, determine the effect of drought stress on two photosystems, and examine the mechanism of acquired drought tolerance. In this study, blue honeysuckle under four levels of simulated field capacity (100%, 85%, 75%, and 65% RH) was grown in split-root pots for drought stress treatment, for measuring the changes in chlorophyll content, photosynthetic characteristics, and leaf chlorophyll fluorescence parameters. The chlorophyll content of each increased under mild stress and decreased under moderate and severe stress. The net photosynthetic rate, transpiration rate, intercellular carbon dioxide concentration, and stomatal conductance of blue honeysuckle decreased with the increase in water stress. However, the water utilization rate and stomatal limit system increased under mild and moderate stress and decreased under severe stress. The maximum fluorescence (Fm), maximum photochemical efficiency, and quantum efficiency of photosystem II decreased with the decrease in soil water content, and the initial fluorescence increased significantly (*p* < 0.01). With the decrease in soil water content, the energy allocation ratio parameters decreased under severe drought stress. The main activity of the unit reaction center parameters first increased and then decreased. ABS/CSm, TRo/CSm, ETo/CSm, and REo/CSm gradually declined. After a comprehensive analysis, the highest scores were obtained under adequate irrigation (CK). Overall, we concluded that the water irrigation system of blue honeysuckle should be considered adequate.

## 1. Introduction

Blue honeysuckle (*Lonicera caerulea* L.) is a newly emerging small perennial berry shrub [1] that is widely distributed in China, Canada, Japan, and Russia [2,3]. It is relatively easy to cultivate and is best grown at high altitudes and in cold climates [4,5]. It contains high levels of anthocyanin, ascorbic acid, and phenolics [6,7,8]. The fruits can be used to make various foods, such as fruit wine, jam, juice, dried fruit, and more. In addition to this, they can also be deeply processed to create additional products like face masks, cosmetics, lipstick, medicines, and other supplementary items [9,10]. There has been a lot of research on the range of its introductions, domestication, variety breeding, processing, and utilization in recent years [11,12,13,14,15]. With increasing attention being paid to it at home and abroad, it is of great significance to study its standardized cultivation mode in order to achieve large-scale popularization; moreover, there is a need to improve water use productivity in driving the standard of berry production [16].

Drought stress, particularly during the anthesis and fruit developmental periods, can reduce yield and fruit quality, as it shortens the life cycle of the fruit and the availability of photosynthates for fruit volume filling [17,18]. It has been reported that the production of photosynthetic material plays an important role in berry growth and yield formation and that the production of biomass and the physiological characteristics of the berries are greatly affected by drought stress [19]. Water deficit affects the transpiration rate, photosynthetic rate, and intercellular carbon dioxide of plants and inhibits plant growth.

Photosynthesis is one of the most important processes that is significantly affected by drought stress [20,21]. The photosynthetic reaction of plant leaves has a light reaction stage and a dark reaction stage. In the dark reaction stage, drought stress can cause changes in the epidermal stomata, mesophyll stomata, and metabolism, and decreases in the leaf net photosynthetic rate and stomatal conductance [22,23,24]. Drought stress also has a certain impact on photosynthetic pigments, decreasing the net photosynthetic rate of plants and the energy produced by photosynthetic pigments [25]. Chlorophyll fluorescence kinetics has become a powerful tool for monitoring and quantifying the behavior and performance of photosynthetic processes [26], mainly on photosynthesis I (PS I) and photosynthesis II (PS II), such as photochemical efficiency and the function and structure of photosynthetic electron transport [27]. In addition, it has been further developed to localize the effect on the PS II architecture and behavior [28,29], for the shape of the O-J-I-P fluorescence transient is sensitive to pressures caused by changes in many environmental conditions [30,31,32,33].

For blue honeysuckle, irrigation is an important factor affecting the growth and development of different parts of blue honeysuckle berry, such as the root system, leaf screen, and bud [11,34,35,36]. Previous studies have shown that the blue honeysuckle critical point for drought tolerance is 50–55% [37]. When plants are subjected to different degrees of water regulation deficit, they have a corresponding drought stress response. Therefore, in this study, we selected 3-year-old blue honeysuckle as a material and performed drought acclimation treatments with different soil moisture contents (with the same irrigation time) at the flowering stage to study the changes in chlorophyll content, chlorophyll fluorescence, and photosynthetic properties using the Li-6400 portable photosynthesis system (LI-COR Inc. Lincoln, NE, USA) and Handy PEA fluorometer (Hansatech Instruments Ltd., King’s Lynn, Norfolk, UK). The results are expected to provide some theoretical support for blue honeysuckle photosynthesis, chlorophyll fluorescence, and drought tolerance.

## 2. Results

### 2.1. Chlorophyll Content

Chlorophyll is the key photosynthetic pigment for solar radiation absorption and chemical energy conversion in plants, and its content is an important index for measuring the physiological and ecological functions of plants [38,39]. Figure 1 shows that chlorophyll a, chlorophyll b, chlorophyll a/b, chlorophyll a + b, total chlorophyll content, and carotene of the uppermost expanded leaves of blue honeysuckle increased and then decreased with increased drought stress. Both varieties had the highest pigment content at a slight deficit adjustment (T1, 85% RH). There was no significant difference in chl a/b, chl (a + b)/(x + c), or chl c/(x + c) for any treatment.

### 2.2. Effects of Drought Stress on Photosynthetic Characteristics

According to the results shown in Figure 2, drought stress changed the photosynthetic characteristics of blue honeysuckle leaves in light response parameters. Under different levels of water drought stress, *Pn*, *G_S_*, *Ci*, and *Tr* significantly decreased, with extremely significant differences expressed (*p* ≤ 0.01). The difference between the T3 treatment group and the control group was the most significant, with reductions of 88.81% (*Pn*), 92.90% (*Gs*), 8.10% (*Ci*), and 92.90% (*Tr*), respectively. WUC and Ls increased first and then decreased under the aggravation of soil drought stress. The T2 and T3 treatments were obviously higher than the CK and T3 treatments. These results suggest that drought stress was not beneficial to increasing the *Pn*, *Gs*, *Ci*, and *Tr*, but was positive for *WUC* and Ls.

### 2.3. Effects of Drought Stress on Chlorophyll Fluorescence Parameters

#### 2.3.1. Raw Fluorescence Rise Kinetics, OJIP Curves, and Relative Variable Fluorescence

The fluorescence transient curves of adequate irrigation (CK), slight deficit adjustment (T1), moderate deficit adjustment (T2), and severe deficit adjustment (T3) are presented in Figure 3A. Clearly, the fluorescence induction transient curves of the four types of drought stress leaves exhibit different influences on a typical polyphasic O-J-I-P shape than those of the control. Comparing the dark-adaptation kinetics of the control leaves, a decrease in the variable fluorescence intensity (Ft) from Fo to Fm in the other treatments was observed, and the OJ phase was gradually suppressed, but the fluorescence kinetics curve from 2 to 30 ms (JP step) was obviously inhibited. Further analysis revealed that the F_0_ of the T1, T2, and T3 treatments increased significantly compared with the control. The change in the F_0_ value is related to the modification of the structure and order of light-harvesting complexes during solar energy absorption. As the water stress increased, the treatment significantly decreased compared with the control; therefore, F_V_/F_O_ decreased due to a lower F_M_ (Figure 3D).

To further analyze the effect of different drought stress treatments on the kinetics OJIP properties, the fluorescence curves double normalized by F0 (20 µs) and FM were presented as relative variable fluorescence V_t_ = (F_t_ − F_0_)/(F_M_ − F_0_) (Figure 3B) and ΔV_t_ (Figure 3C) vs. a logarithmic time scale. Compared with the control, the fluorescence curves of the three drought stress treatments showed apparent and visible changes under normalization (Figure 3C). An analysis of the fluorescence transients revealed that the major effects of the different drought stress treatments for blue honeysuckle leaves occurred in the 0-P phase; therefore, changes were obvious in multiple turnover events of photosynthesis II (0-J, J-I, and I-P transitions), and there were very complex fluctuations during those periods. Because changes in this interval had multiple ups and downs on the effect of aging of PSII on the relative variable fluorescence from 0 to 1000 ms, the changes in the four intervals were amplified separately and analyzed.

#### 2.3.2. L-Band and K-Band

To further evaluate the events reflected in the OK (0.02–0.3 ms), OJ (0.02–2 ms), OI (0.02–30 ms), and IP (30–300 ms) phases, other normalizations and corresponding subtractions (difference kinetics) of the fluorescence rise kinetics curves were also determined. As shown in Figure 4A,B, the fluorescence rise kinetics curves of different treatments were double normalized by O (20 µs) and K-step (300 µs) to show the L-band as W_OK_ = (F_t_ − F_O_)/(F_K_ − F_O_) kinetics (Figure 4A) and plotted with the difference kinetics ΔW_OK_ = W_OK_ (treatment or control) − W_OK_ (control) (Figure 4B) in the liner time scale from 0 to 300 µs. The L-band is an indicator of PSII cell grouping or the energy connection between the antenna and the PSII RCS [40]. T1 and T3 increased the L-band most obviously, and T2 and the control slightly increased the L-band. The W_L_ and ΔW_L_ values of the T3 samples increased significantly, but the F_L_/F_J_ values of the T3-treated samples did not show significant changes compared with the control (Figure 4E). The L-band occurred because, in T3, the DCMU-caused lift in the L-band was due to the increase in the J-step.

To ensure the effects of variable treatments on the K-step, the above diagram (Figure 4C,D,F) was further normalized by double standardization of the O-J phase. The decrease in the K-step is typically attributed to enhanced electron transfer on the PSII donor side [41]. Only the T2 treatment significantly decreased the value of the OEC centers (Figure 4F), corroborating that T3 indeed reinforced the fraction of the active OEC centers.

#### 2.3.3. O-I and IP Phases

The fluorescence kinetics curves were also normalized between the 0-step and I-step (30 ms) as W_OI_ = (F_t_ − F_O_)/(F_I_ − F_O_) (Figure 5A, top) and ΔW_OJ_ = W_(treat)_ – W_(control)_ (Figure 5A, bottom) on a logarithmic time scale. The maximum amplitude of the W_OI_ curve involves information about the pool size of the end electron acceptors on the PSI acceptor side. A smaller amplitude indicates a stronger inhibition effect on the pool size. Under drought stress, each treatment rose to the maximum value at about 300 ms, and the maximum value was observed with the T2 treatment (Figure 5A). ΔW_OI_ was made to show the effects of three drought stress treatments with control on the J-step, but the results were inconsistent with those of ΔVt. The leaf fluorescence parameters of blue honeysuckle had a maximum value at 1.4 ms under slight deficit adjustment (T1, 85% RH), and the minimum value was observed at 100 ms under severe deficit adjustment (T3, 55% RH) (Figure 5B). As shown in Figure 4B, the time point of W_IP_ = 0.5 (half-time of the rise curves) could be used to reflect the reduction rate of the PSI end electron acceptors’ pool. The slight deficit adjustment (T1, about 90 ms), moderate deficit adjustment (T2, about 90 ms), and severe deficit adjustment (T3, about 100 ms) treatments were greater than the adequate irrigation (CK, about 80 ms), suggesting that increased drought stress decreases the reduction rate of PSI upper terminal electrons.

#### 2.3.4. JIP Parameters Estimating the Quantum Yields, Efficiencies, Probabilities, and Performance Index

Table 1 shows that, with the decrease in soil water, the quantum yield and equivalent parameters between each treatment decreased (except φ*_Eo_*, whose trend showed no differences). There were two directions for energy flux absorbed by the PSII single active reaction center: capture (*φ*) and energy dissipation (*Ψ*) [42]. With drought stress intensification, the quantum yields φ_Ro_ of unit ABS and the quantum efficiencies *Ψ*_Eo_ and *Ψ*_Ro_ of unit TR decreased significantly. δ_RO_ is a JIP test parameter expressing the probability that an electron will be transported from the reduced intersystem electron acceptors to the final electron acceptors of PSI. The moderate deficit adjustment treatment (T2) significantly increased δ_RO_ compared with the control and other treatments. PI_ABS_ reflects the overall performance of PSII using an index of PSII functional activity based on PI_ABS_ absorption [43]. As shown in Table 1, the PI_ABS_ value of the severe deficit adjustment treatment (T3) decreased the general performance of PSII more than CK.

#### 2.3.5. Leaf Models Showing the Phenomenological Energy Fluxes per Excited Cross-Section

The phenomenological energy fluxes per excited cross-section (CS) of the mature section of blue honeysuckle leaves under different water stress treatments are presented in Figure 6. As the water deficit stress increased, the approximated absorption flux per CS (ABS/CS_M_), trapped energy flux per CS (TR_0_/CS_M)_, percentage of active/inactive reaction centers (RC_0_/CS_M_), electron transport flux per CS (ET_0_/CS_M_), and dissipated energy flux per CS (DI_0_/CS_M_) decreased.

#### 2.3.6. Pipeline Models of Specific Energy Fluxes per PSll Active Reaction Center

Figure 7 shows the differences between the membrane models of different treatments. After different drought stress treatments, there were no significant differences between adequate irrigation (CK), slight deficit adjustment (T1), moderate deficit adjustment (T2), and severe deficit adjustment (T3) on ABS/RC or TR_0_/RC. However, the ET_0_/RC and RE_0_/RC of these treatments decreased with the decline in irrigation, and DI_0_/RC increased.

### 2.4. Correlation Analysis of Pigment Content with Photosynthetic Characteristics and Minimally Significant Chlorophyll Fluorescence Values

Figure 8 shows the correlation matrices for chlorophyll content with photosynthetic characteristics and chlorophyll fluorescence parameters under different drought stress treatments. Overall, the photosynthetic pigments were correlated with different degrees only in their indexes, while some photosynthetic parameters were correlated with the chlorophyll fluorescence values.

Among the photosynthetic pigments, Chl a and Chl b were correlated with Chl a + b, carotene, and chlorophyll, and the correlation coefficients of the former were 0.98, 0.99, and 1.00, respectivley. The latter was extremely significant, and the correlation coefficients were 1.00, 1.00, and 0.99, respectivley. Chl a/b was significantly positively correlated with chlorophyll (a + b)/(x + c) (0.99) and significantly negatively correlated with chlorophyll c/(x + c) (1.00). Chl (a + b) was significantly positively correlated with carotene (1.00) and chlorophyll (0.99). Carotene was very significantly positively correlated with chlorophyll (1.00). Chl (a + b)/(x + c) was very significantly negatively correlated with chlc/(x + c) (−1.00).

As for the photosynthetic parameters, Pn was positively correlated with Gs (0.99), Tr (0.99), Fm (0.97), Fv (0.98), Fv/Fo (0.99), and PI_ABS_ (0.97). Gs was very significantly positively correlated with Tr (1.00). Ls was very significantly negatively correlated with Ci (−1.00). As for the chlorophyll fluorescence values, Fv/Fm was positively correlated with Vj (0.97), Fm (0.99), and Fv (0.99). Fv/Fo was positively correlated with Fm (1.00), Fv (1.00), and Fv/Fm (0.99). PI_ABS_ was negatively correlated with Vj (−0.97), Fo (−0.96), and Fm (−0.98). PI_ABS_ was positively correlated with Fv/Fm (0.97) and Fm/Fo (0.99).

### 2.5. Principal Component Analysis and Comprehensive Evaluation

Principal component analysis of the photosynthetic parameters of blue honeysuckle leaves after drought regulation was used to obtain the principal component eigenvalue, contribution rate, and cumulative contribution rate (Table 2). Two principal components were selected, and their cumulative contribution rate was greater than 85%. The eigenvalue of the first principal component was 5.66, representing 70.76% of the original information of seven leaf indexes under different drought stress treatments. The eigenvalue of the second principal component was 1.29, which represented 16.08%. The cumulative variance contribution rate of the first two principal components was 86.84%, indicating that these two principal components reflected 86.84% of the original variable information; therefore, the first two principal components were extracted instead of the original eight leaf photosynthetic indicators to evaluate different drought treatments. The evaluation indexes of the different treatments were reduced from the initial seven aspects to two uncorrelated principal components, achieving the purpose of dimensionality reduction.

Since the contribution rate of each main component differed, when the comprehensive evaluation of different drought stress treatments was carried out, the contribution rate of the main component could be combined to better coordinate the emphasis relationship between the main components. The feature vector (Table 3) was used as the weight to construct the expression function formula of the two principal components, as follows:P1 = 0.5846 × X_1_ + 0.9007 × X_2_ − 0.0116 × X_3_ + 0.9737 × X_4_ + 0.9737 × X_5_ + 0.9427 × X_6_ + 0.9738 × X_7_ + 0.8807 × X_8_
P2 = 0.5057 × X_1_ − 0.3507 × X_2_ + 0.9419 × X_3_ + 0.0201 × X_4_ − 0.0125 × X_5_ + 0.1217 × X_6_ − 0.0647 × X_7_ − 0.0317 × X_8._

In the above two expressions, X_1_, X_2_, X_3_, X_4_, X_5_, X_6_, X_7_, and X_8_ are standardized chlorophyll, Gs, WUC, Fm, Fv, Fv/Fm, Fv/Fo, and PI_ABS_, respectively (Table 4). Using the variance contribution rate corresponding to each principal component as the weight, the comprehensive evaluation function was obtained by the linear weighted summation of the principal component score and the corresponding weight. The comprehensive analysis was calculated using the following equation:P = (0.7076/0.8694) × P1+ (0.1608/0.8684) × P2

Based on the principal component comprehensive score model, the comprehensive score and the ranking of the effects of different treatments on leaf photosynthetic parameters were calculated (Table 5). The comprehensive score from high to low was as follows: CK > T1 > T2 > T3.

## 3. Discussion

### 3.1. Effect of Drought Stress on Chlorophyll Content and Photosynthetic Characteristics of Blue Honeysuckle Leaves

Chlorophyll content plays an important role in photosynthesis, and water deficit is an important factor in chlorophyll degradation. Its content can affect the drought resistance of plants to a certain extent [44]. In the process of this study, the content of chlorophyll a, chlorophyll b, and carotenoid in the leaves was highest under slight deficit adjustment treatment (100% RH, T1) (Figure 1). However, in the study of the plantain tree, the chlorophyll content decreased continuously under water stress. The gradient of drought stress we set for this experiment (CK: 100%; T1: 85%; T2: 70%; T3: 55%) was finer than that used by Akhbarfar (2023) [45] (100% field capacity (FC) (D100); 75% FC (D75); and 50% FC (D50): 100%), and we captured the cause of the change in this gradient. In other research about two strawberry cultivars (Kurdistan and Selva) under drought stress (75% (C: control); 50% (S1: mild drought stress); 25% (S2: severe drought stress)), the chlorophyll a and b increased first from 100% to 75%, and then decreased from 75% to 50% on the cultivar of Kurdistan and Selva (Selva was at a significant level) [46], which is consistent with our experimental results. In addition, our research results may indicate that mild drought activates plant protection systems, increasing Chl a and Chl b to maintain normal photosynthesis. However, with the aggravation of drought stress, the formation path of chlorophyll content causes irreversible damage, gradually decreasing the chlorophyll content.

In general, drought stress causes plants to close their stomata, reducing the CO_2_ concentration in the mesophyll, thereby directly inhibiting photosynthesis or inhibiting carbon metabolism, resulting in reduced photosynthesis [47,48]. When soil moisture decreases, the decrease in the net photosynthetic rate (Tr) and stomatal conductance (Gs) may be due to stomatal closure carried out to prevent water loss [49]. Stomatal closure reduces the CO_2_ uptake from the atmosphere into the leaves and limits photosynthesis [50]. Some studies have shown that G_S_ and Tr can be used as parameters to assess plant canopy productivity [50,51]. In this study, the net photosynthetic rate (Pn), stomatal conductance (Gs), intercellular carbon dioxide concentration (Ci), and transpiration rate (Tr) of blue honeysuckle subjected to drought stress decreased compared with the control treatment. This result is consistent with Jin et al. [52], who researched the photosynthetic characteristics of Sargent’s cherry. Leaf water use efficiency (WUC) and stomatal limitation (Ls) depicted inverted ‘V’ patterns, with peaks appearing to have at slight deficit adjustment (85% RH, T1) and moderate deficit adjustment (70% RH, T2), respectively. This may indicate that moderate drought regulation can reduce stomatal conductance and the transpiration rate. In addition, plants may reduce the impact of drought by closing their stomata, reducing the transpiration rate and improving water use efficiency to ensure plant growth. Xing et al. [53] found that, under the condition of moderate nitrogen content, the variation trend in water use efficiency was consistent with our results, while both nitrogen deficiency and nitrogen excess showed a decreasing trend with the increase in water stress. Therefore, the nitrogen content in this study may also be moderate.

### 3.2. The Effect of Drought Stress on the Rapid Chlorophyll Fluorescence Characteristics of Blue Honeysuckle Leaves

Photosynthesis is the physiological basis of plant growth and development. When photosynthesis is inhibited, chloroplast photosystem II is the first to be affected. Understanding the regulatory mechanism of PSII under stress can effectively reflect the survival mode of plants under stress. Plant photosynthesis dynamics have become a research hotspot, and its parameters have been widely used in the study of water stress physiology [54]. In this study, Fm and Fv/Fm declined as drought stress increased. The gradual intensification of drought stress may cause irreversible damage to the internal structure of blue honeysuckle leaves, which is consistent with the results of other studies on the Fv/Fm index [55,56]. Our research further demonstrates that leaf Fv/Fm can be used as an important index to evaluate the resistance of blue honeysuckle to water stress. PI_ABS_ is a comprehensive evaluation index of photosynthetic response efficiency, which is a product of three independent components consisting of φ_Po_, ψ_Eo_, and γ_RC_, reflecting the effect of stress on the photosynthetic structure to some extent (Gao, et al., 2018). PI_ABS_ showed no significant difference between adequate irrigation (100% RH, CK) and moderate deficit adjustment (85% RH, T1), but the value of severe deficit adjustment very significantly decreased compared with moderate deficit adjustment (70% RH, T3) (Table 2). This may indicate that the total photosynthetic performance of the indigo leaves decreased significantly under extreme drought conditions.

Regarding leaf models showing the phenomenological energy fluxes per excited cross-section, many studies have indicated that several sites in PSII are sensitive to various types of environmental stress [57,58,59]. In our study, the decrease in ABS/CS_M_ reflected an increased density of inactive reaction centers in response to drought stress (Figure 4). TRo/CSm and Eto/CSm gradually decreased with increasing water stress concentration on blue honeysuckle leaves (Figure 4) because active reaction centers (RCs) are converted into inactive RCs, which reduces the energy trapping efficiency and electron transport from PSII. Meanwhile, reduced energy trapping and electron transport per excited cross-section (F_M_) has been observed with increasing water stress (Figure 3B), which may indicate that a decreased energy absorption efficiency of PSII may be caused by an increase in inactive RCs. Similar results have been reported in rice leaves when the root system of the plant was treated with different degrees of drought stress [60].

The rapid chlorophyll fluorescence induction kinetics curve accurately reflected absorption, projection, distribution, and dissipation [61,62,63]. Studies have shown that adverse stress conditions affect electron transport in leaves and inhibit Q_A_–Q_B_ of electron transport in photosynthesis [17,64,65]. TRo/RC, ETo/RC, and REo/RC reached a peak value at a slight adjustment of deficit treatment (75–90%, T1) and then decreased, which showed that, after moderate drought stress was triggered, the inactivation or cleavage of the reaction center per unit area of the leaf and the efficiency of the remaining active reaction centers were promoted to better dissipate the energy in the electron transport chain. However, after drought stress reached this critical point, the efficiency of electron transport and energy conversion decreased due to damage to the PSII structure.

Because it has certain limitations for a single index to evaluate plant drought status, principal component analysis was carried out after eliminating the correlation index from the plant chlorophyll content index, photosynthetic index, and main index of chlorophyll fluorescence. We concluded that the comprehensive effect of adequate irrigation (100% RH, CK) was better than that of the other treatments. Overall, adequate irrigation treatment gained the best overall effect.

## 4. Materials and Methods

### 4.1. Plant Materials and Growth Conditions

A pot experiment was carried out to investigate the responses of 2-year-old blue honeysuckle (*Lonicera caerulea* L.) to drought stress treatment. Using small berry germplasm resources, the experiment was conducted in April 2023 in a greenhouse at the Horticultural station of Northeast Agricultural University (E126°43′, N45°44′), located in Harbin, Heilongjiang province, China. To create a microclimate environment that is completely rain-free and basically consistent with the outdoor environment, the greenhouse was opened or closed by humans according to the weather conditions. In April, potted blue honeysuckle ‘A5’ seedlings of good and consistent growth at 50 cm high were placed into the greenhouse for slow seeding. The planting pot basin had an upper diameter of 30 cm, height of 30 cm, and thickness of 0.10 cm. The plastic pots were filled with 16.77 kg of black soil when it had been oven-dried to constant weight, and the maximum soil absolute water content was 25% and the amount of water in the soil at maximum absolute water content was 4.26 kg.

### 4.2. Experiment Design

After the plants were left to acclimate for two weeks in the greenhouse and were well-watered, irrigation was withheld for three plants in each treatment for three days until the relative soil moisture content was 55%, as measured with a YLS16-A moisture test apparatus measuring soil humidity (Halogen tube heating type, Echcomp, Shanghai Tianmei Balance Instrument Co., Ltd., Shanghai province, China). The water stress level was divided into the following four gradients: adequate irrigation (100% RH, 25% water content of soil), slight adjustment (85% RH, 21.25% soil absolute water content), moderate deficit adjustment (70% RH, 17.50% soil absolute water content), and severe deficit adjustment (55% RH, 13.75 soil absolute water content) (Table 6). The plants were then watered with tap water every three days according to the water gradient adhering to the following formula:$$M=\frac{\left(A-B\right)\times D}{C}$$
where *M* is the amount of watering in each treatment; *A* is the set soil absolute water content; *B* is the determination of soil absolute water content; *C* is the maximum soil absolute water content (25%); and *D* is the amount of water under the maximum field capacity (16.77 kg).

After continuous drought treatment repeated 5 times, the corresponding photosynthetic characteristics were determined. Each treatment was carried out with a blue honeysuckle plant as a repeat and repeated 3 times, totaling 12 seedling pots.

### 4.3. Determination of Chlorophyll Content

The protocol of Gao [66] was followed to determine the leaf chlorophyll content one day after drought stress treatment. A total of 0.1 g of leaves from the middle branch was weighted, cut into pieces, and placed in 20 mL of a mixture of absolute ethanol and water (*v*/*v* = 95:5). The uppermost unfolded leaf was incubated in the dark at room temperature (25 °C) for 12 h. After the leaves completely faded, the residue was filtered, and the supernatant was retained. The Bio Tek Synergy multifunctional enzyme marking instrument (Agilent Technologies (China) Inc., Chaoyang district, Beijing, China) was used to measure the absorbance of the sample at 665, 649, and 470 nm. The formats were as follows:Chlorophyll a = 13.95 × A_665_ − 6.88 × A_649_;
Chlorophyll b = 24.96 × A_649_ − 7.32 × A_665_;
Chl a/b = Chlorophyll a/Chlorophyll b;
Chl a + b = Chlorophyll a+ Chlorophyll b;
Carotenoids = (1000 × A_470_ − 3.27Ca − 104Cb)/245;
Total chlorophyll content = Chl (a + b) + carotenoids;
Chl (a + b)/(x + c) = Chl a + b/(Chlorophyll a + Chlorophyll b + Carotenoids);
Car/(x + c) = Carotenoids/Total chlorophyll content.

### 4.4. Photosynthetic Measurements

Photosynthesis was performed using a Li-6400 portable photosynthesis system (LI-COR Inc., Lincoln, NE, USA) from 9:00 to 10:00 a.m. on 18 June 2023. The physiological parameters’ net photosynthetic rate (Pn, μmol/(m^2^·s)), stomatal conductance (Gs, mol/(m^2^·s)), transpiration rate (Tr, mmol/(m^2^·s)), and intercellular CO_2_ concentration (Ci, μmol/mol) were measured in situ on the third or fourth leaf that was fully expanded (counted back from the apex of new shoots). The leaf water use efficiency (WUE) was calculated as WUE = Pn/Tr (mmol/mol). The stomatal limitation (Ls) was calculated as Ls = 1 − Ci/Ca [67]. Three leaves were chosen from each tree, and the test was repeated three times.

### 4.5. Measurement of Chlorophyll Fluorescence Rise Kinetics

Because the light source used by the modulated fluorometer to measure fluorescence is modulated pulse light (high-frequency flash), a fluorescent signal is emitted by the plant. It can be distinguished from the light emitted by the instrument light source, therefore, it can be measured in the presence of background light. The fast chl a fluorescence induction kinetics OJIP curves of the detached intact leaf of blue honeysuckle were determined using a plant efficiency analyzer (Handy PEA fluorometer, Hansatech Instruments Ltd., King’s Lynn, Norfolk, UK).

Before starting the experiment, we cleaned the leaves with distilled water and then wiped them off with paper towels. When we started the experiment. We first clamped the leaves from different treatments at uniform temperatures with the purified water. When clamping the leaves with a dark-adapted clip, the middle hole of the clip was fully filled, avoiding veins, and the leaves were kept in the dark for half an hour with the septum. A continuous red light (680 nm, PAR of 3000 μmol (photon) m^−2^ s^−1^) was provided to induce the fluorescence transient OJIP curve [68]. The following data from the original measurements were used: F_0_ (the fluorescence intensity at 20 μs), F_J_ (the fluorescence intensity at 2 ms), F_I_ (the fluorescence intensity at 30 ms), and the maximal fluorescence intensity F_M_ (equal to F_P_).

We selected important quantum parameters of the electron transport chain from the plant efficiency analyzer, counted the phenomenological flux leaf model by approximated absorption flux per CS(ABS/CSM), trapped energy flux per CS (TR_O_/CSM), electron transport flux per CS (ET_O_/CSM), and dissipated energy flux per CS (DI_O_/CSM), and calculated the special fluxes membrane model from the absorbed photon flux per active PSII (ABS/RC), trapped energy flux per active PSII (TR_O_/RC), electron flux from Q_A_^−^ to the PQ pool per active PSII (ET_O_/RC), and dissipated energy (as heat and fluorescence) flux per active PSII (DI_O_/RC). An extended analysis of OJIP transients was performed by calculating the relative variable fluorescence, as follows: maximum variable fluorescence (F_V_ = Fm − Fo), maximum quantum yield of primary photochemistry (F_V_/F_M_ = φpo), relative variable fluorescence at time 0 to 1000 ms (V_t_ = (Ft − F_O_)/(F_M_ − F_O_))), the fluorescence rise kinetics with double normalization at time 0 to 1000 ms (ΔV = Vt(treat) − Vt(control))), From F_K_ to F_O_ standardized relatively variable fluorescence (W_OK_ = (Ft − F_O_)/(FK − F_O_)), the fluorescence rise kinetics with double normalization at time 0–0.3 ms (ΔW_OK_ = W_OK (treatment)_ − W_OK (control)_), From F_J_ to F_O_ standardized relatively variable fluorescence (W_OJ_ = (Ft − F_O_)/(F_J_ − F_O_)), the fluorescence rise kinetics with double normalization at time 0–2 ms (ΔW_OJ_ = W_OJ (treatment)_ − W_OJ (control)_), F_I_ − F_O_ standardized relatively variable fluorescence (W_OI_ = (Ft − F_O_)/(F_I_ − F_O_)), the fluorescence rise kinetics with double normalization at time 0–30 ms (ΔW_OI_ = W_OJ (treatment)_ − W_OJ (control)_), quantum yield for electron transport (ET) (φEo = ET_o_/ABS = [1 − (F_O_/F_M_)] ψ_0_ = φ_Po_ ψ_0_], quantum yield for reduction in the end electron acceptors at the PSI acceptor side (RE) (φ_Ro_ = RE_o_/ABS = φ_Po_(1 − V_I_) = φ_Po_ φ_Eo_ δ _Ro_), probability that an electron moves further than Q_A_ (at time 0) (Ψ_Eo_ = ET_0_/TR_0_ = [1 − (F_J_/F_M_)]), the efficiency with which a single exciton trapped in the active reaction center drives a single electron from Q_A_− through the electron transport chain to the PSI acceptor side terminal electron acceptor (at time 0) (Ψ_Ro_ = RE_0_/TR_0_ = ψ_Eo_. Δ_Ro_), and the probability that an electron will be transported from the reduced intersystem electron acceptors to final electron acceptors of PSI (δRo = REo/ETo).

The performance index for energy conservation from photons absorbed by PSII to the reduction in intersystem electron acceptors (PI_ABS_) was introduced as follows:$${\mathrm{PI}}_{\mathrm{ABS}}=\frac{\mathrm{RC}}{\mathrm{ABS}}\times \left(\frac{\varphi_{P_{0}}}{1-\varphi_{P_{0}}}\right)\times \left(\frac{\psi_{E_{0}}}{1-\psi_{E_{0}}}\right)$$

The experiment was repeated three times with at least nine repetitions. The raw fluorescence data were transferred using Handy PEA PLUS software (Version: 1.13, Handy PEA fluorometer, Hansatech Instruments Ltd., King’s Lynn, Norfolk, UK) and analyzed using Excel 2010 software. The Chl fluorescence assays were performed without damaging the plant tissue.

### 4.6. Statistical Analysis

The results are shown as the means ± standard error (SE). For statistical analysis, Excel 2010, Origin 2021, IBM SPSS statistics 23 (SPSS, Chicago, IL, USA), and the data processing system 5.0 (DPS, Hanghzhou city, Zhejiang province, China) were used. The differences between the means were determined by Duncan’s multiple range tests, with a significance level of 0.05 and an extreme significance level of 0.01.

## 5. Conclusions

In this experiment, we studied the chlorophyll content, photosynthetic characteristics, and chlorophyll fluorescence characteristics of indigo fruit at the flowering stage under drought stress. In general, during the critical stage of blue honeysuckle fruit growth, the water deficit had a certain degree of influence on plant nutrient production. Blue honeysuckle had certain adaptability with mild drought stress, an increased chlorophyll content (Chl a, Chl b, and carotenoid content) in the leaves, and an improved water utilization rate. Moderate water stress did not affect the initial fluorescence of PSII but reduced photosynthesis by reducing the chloroplast pigment content, maximum photochemical quantum yield of PSII, and potential photochemical efficiency. Severe water stress caused more serious damage to PSII in the leaves of blue honeysuckle. We conclude that an appropriate amount of drought stress (soil relative water content with adequate irrigation (100% RH)) is beneficial for improving the performance of electron and energy transfer in the photosynthetic electron transport chain of blue honeysuckle leaves.

## Figures and Tables

**Figure 1 plants-13-02115-f001:**
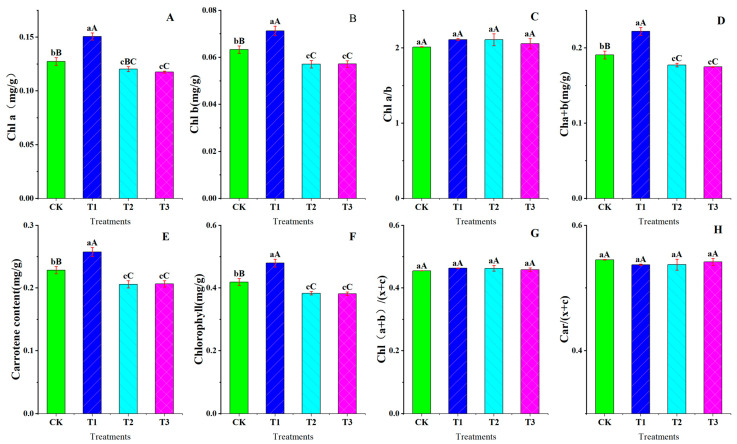
Effect of drought stress on content and relative ratio of photosynthetic pigments in blue honeysuckle leaves under the control treatment (CK), lower drought stress (T1), moderate drought stress (T2), and severe drought stress (T3) in 2023: (**A**) Chlorophyll a content (Chl a); (**B**) Chlorophyll b content (Chl b); (**C**) Chlorophyll a content/Chlorophyll b content (Chla/b); (**D**) Chlorophyll a content + Chlorophyll b content (Chl a + b); (**E**) Carotenoids (Car); (**F**) Total chlorophyll content (Chlorophyll); (**G**) (Chlorophyll a content + Chlorophyll b content)/Total chlorophyll content (Chl (a + b)/(x + c)); (**H**) Carotenoids/Total chlorophyll content (Car/(x + c)). CK: control, 100% RH; T1: 85% RH; T2: 70% RH; T3: 55% RH. RH: relative soil water content. Different letters represent significant differences between treatments at *p* ≤ 0.05 and very significant differences between treatments at *p* ≤ 0.01. Values are the mean ± SE (*n* = 3).

**Figure 2 plants-13-02115-f002:**
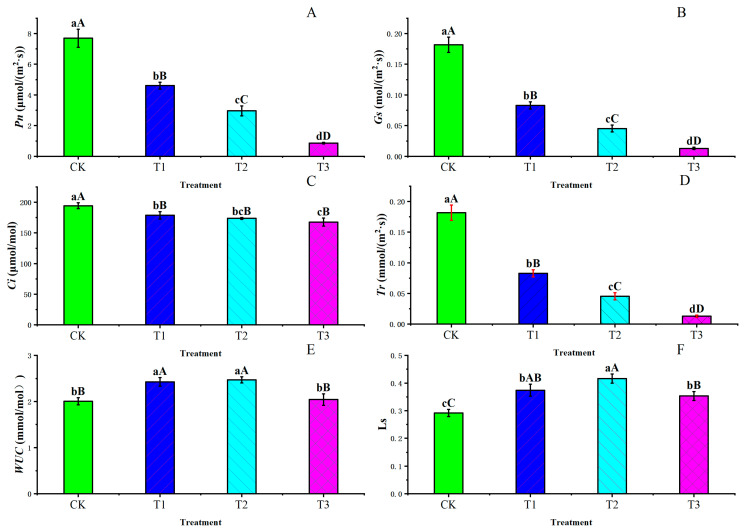
The photosynthetic characteristics under the control treatment (CK), lower drought stress (T1), moderate drought stress (T2), and severe drought stress (T3) in 2023. (**A**): the net photosynthetic rate (*Pn*, μmol/(m2·s)); (**B**): stomatal conductance (*Gs*, mol/(m^2^·s)); (**C**): intercellular CO_2_ concentration (*Ci*, μmol/mol); (**D**): transpiration rate (*Tr*, mmol/(m^2^·s)); (**E**): leaf water use efficiency (*WUE*, (mmol/mol); (**F**): stomatal limitation (Ls). Different letters above the columns indicate a significant difference (*p* ≤ 0.05) and an extremely significant difference (*p* ≤ 0.01). CK: control, 100% RH; T1: 85% RH; T2: 70% RH; T3: 55%; RH: relative soil water content. Values are the mean ± SE (*n* = 3).

**Figure 3 plants-13-02115-f003:**
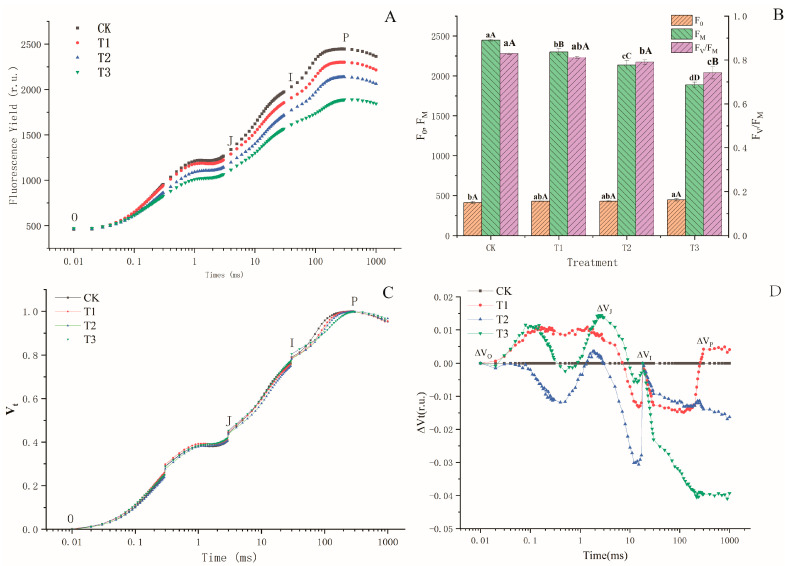
Changes in chl a fluorescence rises kinetics OJIP of blue honeysuckle leaves after drought stress treatment. (**A**) Original data on a logarithmic time scale without any normalization. (**B**) Effects of different drought stress treatments on F_0_, F_M_ and Fv/F_M_ values. (**C**) Fluorescence rise kinetics normalized by F_0_ and F_M_ as V_t_ = (F_t_ − F_0_)/(F_M_ − F_0_). (**D**) The effect of aging PSII on the relative variable fluorescence [ΔV = ((V_t(treat)_ − V_t(control)_))/V_t(control)_)] of the studied leaves. Each curve represents the average of three independent experiments of nine repetitions. CK: control, 100% RH; T1: 85% RH; 70% RH; T3: 55%. RH: relative soil water content. Values are the mean ± SE (*n* = 3). In (**B**), the uppercase letters represent the extreme significance level and the lowercase letters represent the significance level.

**Figure 4 plants-13-02115-f004:**
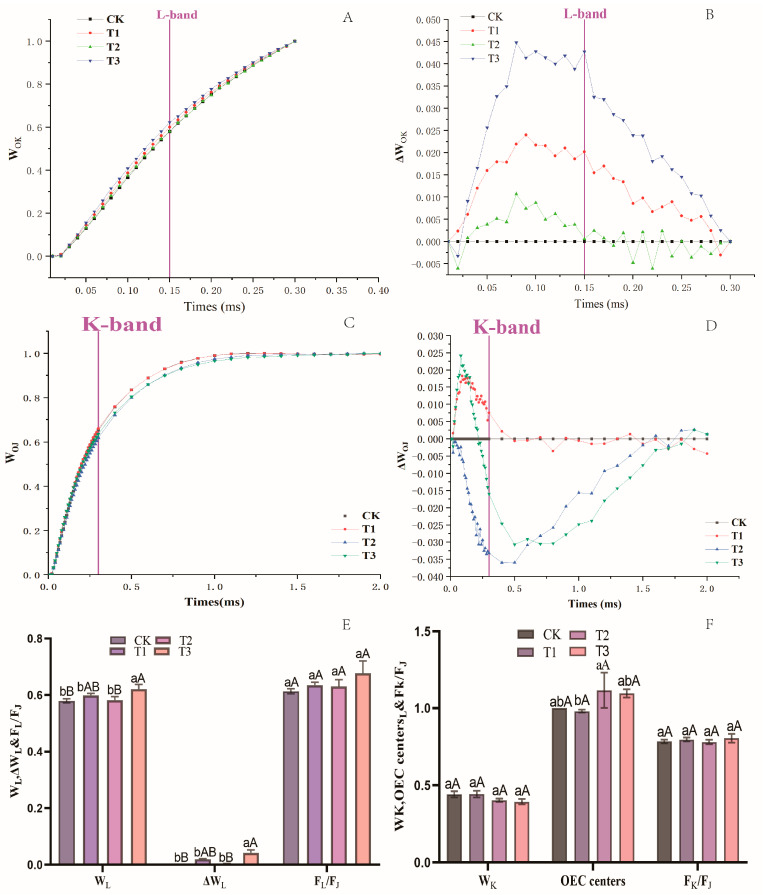
Different normalizations of the fluorescence rise kinetics OJIP curves in a linear time scale from 0 to 300 µs under drought stress and control conditions on blue honeysuckle leaves. (**A**) Fluorescence rise kinetics normalized by F_O_ and F_K_ as W_OK_ = (Ft − F_O_)/(F_K_ − F_O_). (**B**) Difference kinetics ΔW_OK_ = W_OK (treatment)_ – W_OK (control)_. (**C**) Fluorescence rise kinetics normalized by F_O_ and F_J_ as W_OJ_ = (Ft − F_O_)/(F_J_ − F_O_). (**D**) Difference kinetics ΔW_OJ_ = W_OJ (treatment)_ – W_OJ (control)_. (**E**) Values of W_L_, ΔW_L_, and F_L_/F_J_. (**F**) Values of W_K_, OEC centers, and F_K_/F_J_. Each curve is the average of nine measurements. CK: control, 100% RH; T1: 85% RH; T2: 70% RH; T3: 55% RH. RH: relative soil water content. Values are the mean ± SE (*n* = 3). In (**E**,**F**), the uppercase letters represent the extreme significance level and the lowercase letters represent the significance level.

**Figure 5 plants-13-02115-f005:**
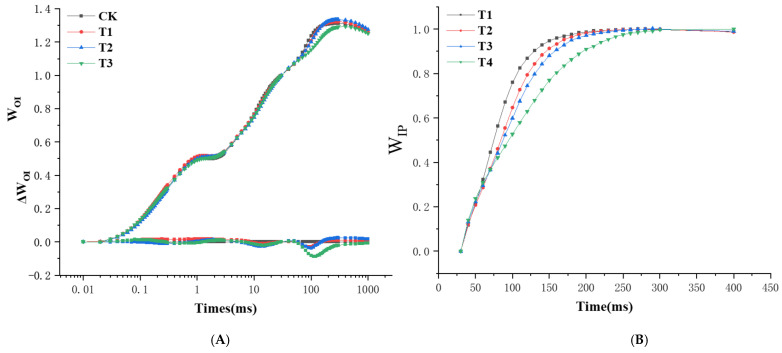
Changes in the fluorescence O-I phase in blue honeysuckle under different drought stress treatments. (**A**) Variable fluorescence between steps O and I as W_OJ_ = (Ft − F_O_)/(F_J_ − F_O_) (top) and ΔW_OJ_ = W(treat) − W (control) (bottom). (**B**) Fluorescence rise kinetics curves normalized by F_I_ and F_P_ (F_M_) as W_IP =_ (Ft–F_I_)/(Fp – F_I_). (D) Probability that an electron will be transported from the reduced intersystem electron acceptors to the final electron acceptors of PSI, δRo. Each curve is the average of nine measurements. CK: control, 100% RH; T1: 85% RH; T2: 70% RH; T3: 55% RH. RH: relative soil water content. Values are the mean ± SE (*n* = 3).

**Figure 6 plants-13-02115-f006:**
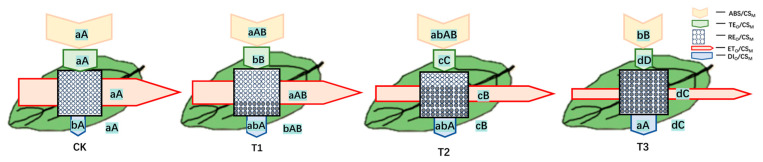
Leaf models showing the phenomenological energy fluxes per excited cross-section (CS) of the leaves of blue honeysuckle under different drought stress treatments during the maturation period. Each relative value of the measured parameters is the mean (n = 9), and the width of each arrow corresponds to the intensity of the flux. ABS/CS_M_: approximated absorption flux per CS; TR_0_/CS_M_: trapped energy flux per CS; RC_0_/CS_M_: percentage of active/inactive reaction centers; ET_0_/CS_M_: electron transport flux per CS; DI_0_/CS_M_: dissipated energy flux per CS. The white circles inscribed in squares represent reduced QA reaction centers (action), and the black circles represent nonreducing QA reaction centers (inaction). Under adequate irrigation, 100% of the active reaction centers responded, with the highest mean value observed in the reference. Means followed by different letters for each parameter are significantly different (*p* < 0.05) or extremely significantly different (*p* < 0.01) according to Duncan’s test. Letters are inscribed into arrows, except for RC_0_/CS_M_, for which they are placed in a box in the lower right corner of the square with circles. Values are the mean ± SE (*n* = 3).

**Figure 7 plants-13-02115-f007:**
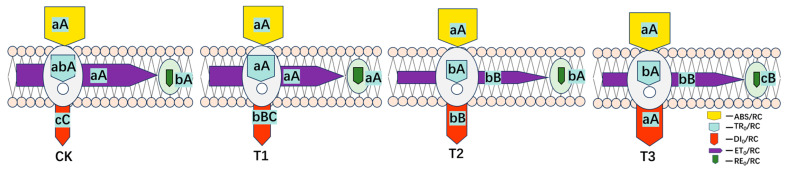
Pipeline models of specific energy fluxes per active PSII reaction center (membrane/specific model) for blue honeysuckle under different drought stress treatments. ABS/RC: Absorbed photon flux per active PSII; TRo/RC: Trapped energy flux per active PSII; DIo/RC: Dissipated energy (as heat and fluorescence) flux per active PSII; ETo/RC: Electron flux from QA−to the PQ pool per active PSII; REo/RC: Electron flux from QA− to the final electron acceptors of PSI per active PSII. CK: control, 100% RH; T1: 85% RH; T2: 70% RH; T3: 55%. RH: relative soil water content. Means followed by different letters for each parameter are significantly different (*p* < 0.05) or extremely significantly different (*p* < 0.01) according to Duncan’s test. Letters are inscribed into arrows. Values are the mean ± SE (*n* = 3).

**Figure 8 plants-13-02115-f008:**
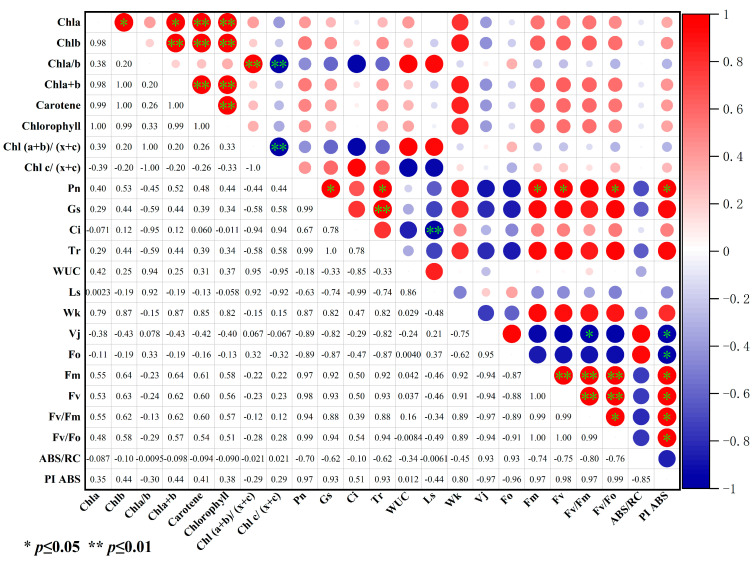
Correlation analysis of pigment content with photosynthetic characteristics, W_K_, V_J_, Fv/Fm, and PI_ABS_. Chl a: chlorophyll a content; chl b: chlorophyll b content; cha/b: chlorophyll a content/chlorophyll b content; Chl a + b: the sum of chlorophyll a content and chlorophyll b content; carotene: total carotenoid concentration; chlorophyll: total chlorophyll; Chl (a + b)/(x + c): (chlorophyll a content + chlorophyll b content)/total chlorophyll content; Chl c/(x + c): total carotenoid concentration/total chlorophyll content; *Pn*: net photosynthetic rate; *Gs*: stomatal conductance; *Ci*: intercellular carbon dioxide concentration; *Tr*: transpiration rate; WUC: leaf water use efficiency; Ls: stomatal limit value; W_K_: normalized K point (0.3 ms) variable fluorescence; V_j_: standardized variable fluorescence at J-point (2 ms); Fv/Fm: maximal photochemical efficiency of PSII; PI_ABS_: performance index.

**Table 1 plants-13-02115-t001:** Mean squares from the analysis of variance (ANOVA) of quantum yield, efficiency/probability, and performance index of blue honeysuckle leaves under different drought stress treatments. φ_Eo_: quantum yield for electron transport (ET); φ_Ro_: quantum yield for reduction in the end electron acceptors at the PSI acceptor side (RE); ψ_Eο_: probability that an electron moves further than QA (at time 0); Ψ_Ro_: efficiency with which a single exciton trapped in the active reaction center drives a single electron from QA through the electron transport chain to the PSI acceptor side terminal electron acceptor (at time 0); δ_Ro_: probability that an electron will be transported from the reduced intersystem electron acceptors to the final electron acceptors of PSI (RE); PI_ABS_: performance index (potential) for energy conservation from photons absorbed by PSII to the reduction in intersystem electron acceptors. CK: control, 100% RH; T1: 85% RH; T2: 70% RH; T3: 55% RH. RH: relative soil water content. Different letters represent significant differences between treatments at *p* ≤ 0.05, determined by Duncan’s test. Values are the mean ± SE (*n* = 9).

Treatment	Quantum Yields and Efficiencies/Probabilities					Performance Index (PI_ABS_)
	φ_Eo_	φ_Ro_	Ψ_Eo_	Ψ_Ro_	δ_Ro_	
CK	0.5 ± 0.02 aA	0.19 ± 0 aA	0.6 ± 0 aA	0.23 ± 0 abAB	0.23 ± 0 abAB	2.79 ± 0.28 aA
T1	0.48 ± 0.01 aA	0.19 ± 0 aA	0.6 ± 0.01 abA	0.24 ± 0.01 aAB	0.24 ± 0.01 aAB	2.37 ± 0.23 aAB
T2	0.47 ± 0 aA	0.2 ± 0.01 aA	0.6 ± 0.01 abA	0.25 ± 0.01 aA	0.25 ± 0.01 aA	2.31 ± 0.32 aAB
T3	0.43 ± 0.09 aA	0.16 ± 0 bB	0.58 ± 0.01 bA	0.22 ± 0.01 bB	0.22 ± 0.01 bB	1.79 ± 0.14 bB

**Table 2 plants-13-02115-t002:** Eigenvalue and contribution rate of principal component analysis leaf photosynthetic parameters under different drought regulation treatments.

Composition	Eigenvalue	Variance Contribution Rate (%)	Cumulative Variance Contribution Rate (%)
1	5.66	70.76	70.76
2	1.29	16.08	86.84
3	0.62	7.75	94.59
4	0.22	2.73	97.32
5	0.12	1.47	98.79
6	0.07	0.93	99.71
7	0.02	0.27	99.98
8	0.00	0.02	100.00

**Table 3 plants-13-02115-t003:** Principal component characteristic vector of the leaf photosynthetic parameter index. Chlorophyll: chlorophyll content; Gs: stomatal conductance; WUE: water use efficiency; Fm: maximal fluorescence; Fv: variable fluorescence; Fv/Fm: maximal photochemical efficiency of PSII; Fv/Fo: PSII functional activity index standardized for ABS absorption; PI_ABS_: performance index.

Photosynthetic Indicators	Feature Vectors	
	1	2
chlorophyll	0.5846	0.5057
Gs	0.9007	−0.3507
WUC	−0.0116	0.9419
Fm	0.9737	0.0201
Fv	0.9731	−0.0125
Fv/Fm	0.9427	0.1217
Fv/Fo	0.9738	−0.0647
PI ABS	0.8807	−0.0317

**Table 4 plants-13-02115-t004:** Standardized data of the leaf photosynthetic parameter index. Chlorophyll: total chlorophyll content; Gs: stomatal conductance; WUE: water use efficiency; Fm: maximal fluorescence; Fv: variable fluorescence; Fv/Fm: maximal photochemical efficiency of PSII; Fv/Fo: PSII functional activity index standardized for ABS absorption; PIABS: performance index. CK: control, 100% RH; T1: 85% RH; T2 70% RH; T3: 55% RH. RH: relative soil water content.

Treatment	Chlorophyll	Gs	WUC	Fm	Fv	Fv/Fm	Fv/Fo	PI_ABS_
CK	0.0774	1.5190	−0.9774	1.0963	1.1056	0.9522	1.1880	1.0992
T1	1.5086	0.0340	0.8040	0.4689	0.4454	0.4684	0.3473	0.1296
T2	−0.7709	−0.5326	0.9936	−0.2497	−0.2366	−0.0668	−0.2233	−0.0007
T3	−0.8150	−1.0204	−0.8201	−1.3155	−1.3144	−1.3538	−1.3120	−1.2281

**Table 5 plants-13-02115-t005:** Principal component analysis score, comprehensive score, and rank of photosynthetic parameters under different drought stress treatments. P1, first principal component score; P2: second principal component score; P: synthesis score. CK: control, 100% RH; T1: 85% RH; T2 70% RH; T3: 55% RH. RH: relative soil water content.

Treatment	P1	P2	P	Rank
CK	6.5905	−1.4018	5.1105	1
T1	2.6871	1.5424	2.4752	2
T2	−1.6963	0.7371	−1.2457	3
T3	−7.5813	−0.8777	−6.3400	4

**Table 6 plants-13-02115-t006:** Drought acclimation treatments for ‘A5’ blue honeysuckle roots.

Treatment	Period of Flowering	Level of Deficit Adjustment
CK	100% RH	Adequate irrigation
T1	85% RH	Slight deficit adjustment
T2	70% RH	Moderate deficit adjustment
T3	55% RH	Severe deficit adjustment

## Data Availability

The data presented in this study are available on request from the corresponding author due to data supporting the findings of this study are available within the article.

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
