# Peer review of "Effects of Drought Stress on Photosynthesis and Chlorophyll Fluorescence in Blue Honeysuckle"

_plants, 2024, doi:10.3390/plants13152115_

Round 1

Reviewer 1 Report

Comments and Suggestions for Authors

There are minor details, please review the observations in the manuscript

Author Response

Comments 1: lines 85 and 86.  equations not described in materials and methods

Response 1:  I agree with this comment. Therefore, I have added the formulate about the Chl a/b, Chl (a+b), Chl (a+b)/ (x+c), and Car / (x+c). this part was added at line 483, 484, 487 and 488 with red mark.

Comments 2: ‘2.2. Effects of Drought Stress on Photosynthetic Characteristics’ This part results require more analysis(line 97)

Response 2: thank you for pointing this out. Therefore, In order to better describe the result of this part, I re-analyzed it in 101-109 with red mark as: According to the Fig. 2 results, drought stress changed the photosynthetic characteristics of blue honeysuckle leaves in light response parameters Under different levels of water drought stress, Pn, GS, Ci, and Tr significantly decreased, with extremely significant differences (P ≤ 0.01). The difference between T3 treatment group and control group was the most significant, with a reduction of 88.81% (Pn), 92.90% (Gs), 8.10% (Ci)and 92.90% (Tr), respectively. WUC and Ls increased first and then decreased under the aggravation of soil drought stress. The T2 and T3 treatments were obviously higher than the CK and T3 treatments. These results suggest at drought stress was not beneficial to increasing the Pn, Gs, Ci and Tr, but positive for the those WUC and Ls.

Comments3: Vt = (Ft − F0) / (FM− F0) (Fig. 2B) and ΔVt (Fig. 2C)

 it's right?

2c stomatal conductance

3c internal concentration

Response 3:  I agree with this comment. Therefore, I have examined the problem carefully, and in light of the actual situation, changed 'Vt = (Ft − F0)/(FM− F0) (Fig. 2B) and ΔVt (Fig. 2C)' to 'Vt = (Ft − F0)/(FM− F0) (Fig. 3B) and ΔVt (Fig. 3C)' in the line 126 with red mark.

Comments 4: ‘the PIABS value of the severe deficit adjustment treatment decreased the general perfor-mance of PSII.’  regarding CK. not all treatments

Response 4: Yes, I agree with your suggestion. I have changed the ‘the PIABS value of the severe deficit adjustment treatment decreased the general perfor-mance of PSII.’ into ‘the PIABS value of the severe deficit adjustment treatment (T3)decreased the general performance of PSII than CK.’

Comments 5: ‘net photosynthetic rate (Tr)’- ‘Tr’ Pn??, clarify if it is photosynthesis or transpiration

Response 5: thank you for pointing out the question. After careful analysis, it's the transpiration rate (Tr) with red mark.

Comments 6: ‘where C is the amount of watering in each treatment; A is the set soil field capacity; B is the determination of soil field water capacity; D is the amount of water under the maximum field capacity (3.8 kg); and E is the maximum field capacity (100%)’.

Why was this formula used to estimate the amount of watering? and it was not made by available moisture

Response 6: thank you for pointing out it. Because our experiment was set with bonsai in greenhouse. In order to maintain the set soil water content when watering, we need to measure the corresponding soil water content every time we water, and make water supplement according to the soil water content.  In order to better describe the watering basis, I have made appropriate additions in the two parts of plant materials and growth conditions and experimental design. In the one hand, in terms of materials and growing environment, I have added the maximum soil water content and the soil water content at treatment time in line of 440 - 443: The plastic pots were filled with 16.77 kg of black soil when it was oven-dried to constant weight, and the maximum soil absolute water content is 25%, the amount of water in the soil at maximum absolute water content is 4.26kg.

 On the other hand, In the experiment design part, I have added the corresponding absolute soil water content after each setting of relative soil water content in the line of 447-460 : After the plants were left to acclimate for two weeks in the greenhouse and well-watered, irrigation was withheld for three plants in each treatment for three days until the relative soil moisture content was 55% as measured by a YLS16-A moisture test apparatus measuring soil humidity (Halogen tube heating type, Echcomp, Shanghai Tianmei Balance Instrument Co., Ltd., Shanghai province, China). This water stress level was divided into four gradients: adequate irrigation (100% RH, 25% water content of soil), slight adjustment (85% RH, 21.25% soil absolute water content), moderate deficit adjustment (70% RH, 17.50% soil absolute water content), and severe deficit adjustment (55% RH, 13.75 soil absolute water content) (Table 6). Plants were then watered with tap water every three days according to the water gradient following the formula:

M = (A-B)*D/E

Where M is the amount of watering in each treatment; A is the set soil absolute water content; B is the determination of soil absolute water content; C is the maximum soil absolute water content (25%); and D is the amount of water under the maximum field capacity (16.77kg).

Reviewer 2 Report

Comments and Suggestions for Authors

The presented manuscript concerns the effect of drought stress on photosynthesis parameters and chlorophyll content.

I have  a few comments to the authors:

1.              The authors should supplement the Introduction Section with information about Blue Honeysuckle, especially its use.

2.              Line 34 – misspelling

3.              Figures 1,2 are poor quality and should be corrected

4.              Figure 1, no exact description of what the x-axis represents

5.              Line 146 - It is not known what the abbreviations Ok, OJ, Ol mean

6.              Line 451 - No information about what kind of water the plants were watered with (e.g. tap, distilled)

7.              There is no information about how long after the induction of drought stress the photosynthetic parameters were measured.

Author Response

Comments 1: The authors should supplement the Introduction Section with information about Blue Honeysuckle, especially its use.

Response 1: I agree with this suggestion. Therefore, I have I have added the description of the use of the blue honeysuckle in line 39-41 with resd mark: Fruits can be used to make various foods such as fruit wine, jam, juice, dried fruit, and more. In addition to this, they can also be deeply processed to create additional products like face masks, cosmetics, lipstick, medicines, and other supplementary items.

Comments 2: Line 34 – misspelling

Response 2: thank you for pointing out it, but I found the 34 line was ‘I introduction’ and could not find where was misspelling. Could you please point out clearly which word is misspelled again?

Comments 3: Figures 1,2 are poor quality and should be corrected

Response 3: Agree. I have modified the figures 1 and 2 with redraw picture. You can see the revised picture before line 87 and in line 104 of my re-uploaded revision with red mark.

Comment 4: Figure 1, no exact description of what the x-axis represents

Response 4: Yes, I agree. Therefore, I have I have revised the description in line 87 – 96 with red mark: ‘Figure 1. Effect of drought stress on content and relative ratio of photosynthetic pigments in blue honeysuckle leaves under the control treatment (CK), lower drought stress (T1), moderate drought (T2), and severe drought stress (T3) in 2023: (A) Chlorophyll a content (Chl a); (B) Chlo-rophyll b content (Chl b); (C) Chlorophyll a content /Chlorophyll b content (Chla/b); (D) Chlorophyll a content + Chlorophyll b content (chl a+b);  (E) Carotenoids (Car); (F) Total chlorophyll content (Chlorophyll); (G) (Chlorophyll a content + Chlorophyll b content) /Total chlorophyll content (Chl (a+b)/(x+c)); (H)  Carotenoids/ Total chlorophyll content (Car/(x+c)). CK: control, 100% RH; T1: 85% RH; T2: 70% RH; T3: 55% RH. RH: relative soil water content. Different letters represent significant differences between treatments at P ≤ 0.05 and very significant differences between treatments at P ≤ 0.01. Values are the mean ± SE (n = 3).’

Comment 5: Line 146 - It is not known what the abbreviations Ok, OJ, Ol means

Response5: Ok, thank you for pointing out it. Because I have described the time nodes of O, J, I and P in lines 506-508 of the manuscript, so I have not described them in this line. In order to better understand the article, I have made some modifications in this line 147- 149 with red mark: To further evaluate the events reflected in the OK (0.02 − 0.3 ms), OJ (0.02 − 2ms), OI (0.02 − 30 ms), and IP (30 − 300 ms) phases, other normalizations and corresponding sub-tractions (difference kinetics) of the fluorescence rise kinetics curves were also determined.

Comment 6: Line 451 - No information about what kind of water the plants were watered with (e.g. tap, distilled)

Response 6: We agree with this comment. I have changed it to the ‘lants were then watered with tap water every three days according to the water gradient following the formula’ in the line 454-455, by the red mark.

Comment 7: There is no information about how long after the induction of drought stress the photosynthetic parameters were measured.

Response 7: Thank you for pointing out it. About your puzzled with there is no information about how long after the induction of drought stress the photosynthetic parameters were measured, I have I have previously described this in lines 463-467 of the manuscript ‘After continuous drought treatment five times, the corresponding photosynthetic characteristics were determined’, and now I will mark this part with a red mark.